# Aggregation Characteristics of Tau Phosphorylated by Various Kinases as Observed by Quantum Dot Fluorescence Imaging

**DOI:** 10.3390/ijms262010122

**Published:** 2025-10-17

**Authors:** Eisuke Ishibashi, Koki Araya, Kota Nakamura, Keiya Shimamori, Koji Uwai, Masahiro Kuragano, Kiyotaka Tokuraku

**Affiliations:** Graduate School of Engineering, Muroran Institute of Technology, Muroran 050-8585, Japan; eimaru1616@gmail.com (E.I.); koki.a.105.2001@gmail.com (K.A.); x9pgeavt@gmail.com (K.N.); simauma8476@gmail.com (K.S.); uwai@muroran-it.ac.jp (K.U.); gano@muroran-it.ac.jp (M.K.)

**Keywords:** Cdk5/p25, GSK3*β*, kinase, MARK4, p38*α*, tau, tau fibrils, quantum dots

## Abstract

This study focused on the abnormal phosphorylation of tau and its aggregation process, characteristic of Alzheimer’s disease, and aimed to compare the morphology and formation process of phosphorylated tau aggregates produced by four kinases: Cdk5/p25, GSK3*β*, MARK4, and p38*α*. Using quantum dots for 2D and 3D structural analysis, tau aggregates were confirmed in non-phosphorylated tau (non p-tau), as well as tau phosphorylated by GSK3*β* and MARK4. Aggregation initiation times were observed around 72 h for non-p-tau, and around 96 h for GSK3*β* and MARK4 phosphorylated tau. The thickness of non-p-tau aggregates was approximately 11 μm, while GSK3*β* aggregates were significantly thicker (13 μm) and exhibited increased density. TEM analysis suggested that tau forming wavy filaments was less prone to forming large aggregates. ThT assays and CD spectra showed an increased *β*-sheet structure for all kinases. Non-p-tau and GSK3*β* exhibited an increased right-twisted *β*-sheet structure, while Cdk5/p25, MARK4, and p38*α* showed an increased left-twisted *β*-sheet structure. The direct correlation between kinase activity and tau aggregate morphology revealed in this study provides a potential mechanistic basis for understanding disease heterogeneity and establishing novel therapeutic targets for AD specifically or for other neurodegenerative diseases as well.

## 1. Introduction

Alzheimer’s disease (AD) is the primary cause of dementia and cognitive impairment in the elderly [1,2]. This disease causes progressive neuronal dysfunction, leading to neuronal atrophy and death [3].

Tau, a microtubule-associated protein, belongs to the Microtubule Associating Proteins family, is primarily expressed in neurons, and is localized to axons. Tau possesses functions that promote microtubule polymerization and stabilization [4], playing a crucial role in the formation of neural networks. In the adult human brain, six major tau isoforms are generated through mRNA splicing [5].

The accumulation of tau in the brain correlates more directly with the progression of AD than A*β* does [6]. Neurofibrillary tangles (NFTs), which are pathologically confirmed in the brains of AD patients, are composed of polyhelical fibrils (PHFs) formed by tau [7]. NFTs are observed not only in AD but also in neurodegenerative diseases referred to as tauopathies. It is believed that tau undergoes excessive post-translational modifications (PTMs), such as phosphorylation, making it prone to aggregation [5,8]. The amyloid cascade hypothesis, a leading theory on AD pathogenesis, proposes that abnormal increases and aggregation of A*β* occur first, subsequently triggering excessive tau phosphorylation and NFT formation [9]. However, recent studies suggest that subcortical tauopathy begins before A*β* deposition, followed independently by A*β* accumulation. This implies tau’s involvement in initiating the AD process, further highlighting its significance [6,10].

The longest tau in the adult brain contains 85 serine (Ser) or threonine (Thr) residues and tyrosine residues. Consequently, approximately 20% of tau may be phosphorylated, with proline-rich regions being particularly susceptible. Over 31 phosphorylation sites have been identified to date, and 86 phosphorylation phenomena have been reported, although it remains unclear whether these are specific to humans. Only 45 phosphorylation phenomena have been reported in tauopathies [5,6,11]. In vitro, tau serves as a substrate for numerous kinases, but only a very small number of kinases actually phosphorylate tau in vivo [12,13]. Furthermore, the phosphorylation sites differ depending on the kinase, and these sites may contribute to the pathology of AD [14,15].

The following four kinases are commonly cited as phosphorylating tau. Phosphorylation of tau by cyclin-dependent kinase (Cdk) 5/p35 is considered a physiological process, while phosphorylation by Cdk5/p25 is regarded as a pathological process [13,16,17]. Cdk5/p25 accumulates in AD brains and is known to possess higher tau phosphorylation capacity than Cdk5 activated by p35. Consequently, it is considered a key kinase in excessive tau phosphorylation [5,18]. p25 is generated when the full-length p35 is cleaved by calpain, a calcium-dependent protease [16,17]. Being more stable than p35, it activates Cdk5 for a longer duration.

Microtubule affinity regulatory kinase (MARK) regulates tau phosphorylation in cultured cells [19] and has been reported to colocalize with the tau Ser262 epitope within NFTs in the brains of AD patients [20]. It is primarily expressed in the brain and selectively phosphorylates the KXGS motif within the tau microtubule-binding domain (MBD) repeat region, as well as other microtubule-associated proteins. Specifically, it has been suggested that phosphorylation of the tau KXGS motif by MARK4 is necessary for neurite outgrowth [5,21,22].

Glycogen synthase kinase *β* binds to microtubules and becomes glycogen synthase kinase 3*β* (GSK3*β*) [23], and its overexpression in cells dramatically increases tau phosphorylation at multiple sites [24].

p38*α* (SAPK2A) is a member of the p38 mitogen-activated protein kinase (MAPK) family, activated by various environmental stresses and inflammatory cytokines. It is highly expressed in brain regions, with neurons being the primary cells expressing p38*α*, involved in both intracellular and extracellular signaling [25,26]. It has been suggested that p38*α* drives a mechanism of interdependence where the initial phosphorylation of tau regulates phosphorylation at other sites, contributing to widespread tau phosphorylation. Notably, the master site Thr181 may function as both a diagnostic and therapeutic target, holding promise for application as an AD biomarker [27].

Furthermore, heparin is currently used as a polyanionic cofactor to induce tau filament formation in vitro. The biological validity of heparin-induced fibrillar structures as a cofactor-induced fibrillation model for tau aggregation in vitro has been questioned [28,29]. Indeed, cryo-electron microscopy of heparin-induced filaments of the longest tau isoform revealed structural differences compared to tau filaments extracted from the brains of human patients [29]. Heparin-induced tau fibrillation has been widely used to screen small molecules as tau aggregation inhibitors, but it may generate false findings due to electrostatic interactions between small molecules and heparin. Therefore, we avoided using heparin during the stage of aggregating phosphorylated tau and ensured sufficient incubation time, observing the aggregation forms of phosphorylated tau using various techniques. As a distinctive observation method, we previously succeeded in labeling tau with quantum dots (QDs) and observing the shapes of aggregates using an inverted fluorescence microscope [30]. QDs possess characteristics such as long-term stability and chemical and physical stability. Furthermore, being nanoscale in size, they emit multicolor fluorescence upon excitation with a single wavelength [31]. Using this technique, we observed the shape of aggregates in two dimensions. We also performed three-dimensional observations of aggregate shapes using confocal laser microscopy.

The accumulation of phosphorylated tau aggregates in the brain promotes the progression of AD. Furthermore, numerous reports have documented that tau is phosphorylated by various kinases [26,27,28]. However, conventional studies have yet to elucidate the morphology of phosphorylated tau aggregates or their formation process. Furthermore, no previous studies have compared how the shape and formation process of phosphorylated tau aggregates are characterized by 3D analysis using multiple kinases representative of AD. In this study, we successfully observed differences in aggregate shape and formation process using QD nanoprobes and other observation methods, employing four kinases: Cdk5/p25, GSK3*β*, MARK4, and p38*α*.

Among the four types of kinases, we analyzed differences in the shape and formation process of tau protein aggregates phosphorylated by GSK3*β* and MARK4, and determined that the morphology of tau changes depending on the type of kinase. This discovery suggests that phosphorylated tau, implicated in AD pathogenesis, may be phosphorylated by specific kinases to exert pathological effects, providing a crucial clue for elucidating the disease mechanism.

## 2. Results

### 2.1. Confirmation of Phosphorylation at Major Tau Phosphorylation Sites by Kinases

We first confirmed whether the major phosphorylation sites on the mouse MBD tau fragment (Figure 1A) were phosphorylated by each kinase. As previously reported, we purified the tau MBD fragment using recombinant *Escherichia coli* and confirmed its expression by SDS-PAGE (Appendix A, Section 4) [32,33,34]. Next, to confirm whether this mouse MBD tau fragment phosphorylates the major phosphorylation sites in each kinase, we performed Phos-tag [35] SDS-PAGE and Western blotting. To phosphorylate the MBD tau fragment, 20 μM tau was incubated with each kinase (5 μg/mL Cdk5/p25, 20 μg/mL GSK3*β*, 20 μg/mL MARK4, 10 μg/mL p38*α*) in 20 mM MOPS (pH 6.8), 1 mM MgCl_2_, and 4 mM ATP buffer at 37 °C for 24 h.

Phos-tag SDS-PAGE confirmed phosphorylation saturation at 24 h for all kinases (Appendix A). Western blots were performed using antibodies targeting the major phosphorylation sites: AT8 targeting phosphorylated Ser202/Thr205 (the primary site for Cdk5/p25 and GSK3*β*), an antibody targeting phosphorylated Ser262 (the primary site for MARK4), and an antibody targeting phosphorylated S396 (the primary site for GSK3*β* and p38*α*) (Figure 1A,B) were used to perform Western blots on tau phosphorylation sites.

Contrary to our expectations, the Western blot results showed increased phosphorylation levels not only for Cdk5/p25 and GSK3*β* but also for p38*α* with the AT8 antibody. As expected, the Ser262 antibody revealed increased phosphorylation levels for MARK4. The S396 antibody confirmed increased phosphorylation levels not only for GSK3*β* and p38*α* but also for Cdk5/p25 (Appendix A, Figure 1C, Table 1). This demonstrated that the phosphorylation buffer used in this study enabled the generation of mouse MBD tau fragments with saturated phosphorylation after incubation at 37 °C for 24 h.

### 2.2. Effect of Tau Phosphorylation Buffer on Tau Aggregation

Next, we investigated whether the phosphorylation buffer used to phosphorylate tau affects tau aggregation. We prepared QD-tau labeled by cross-linking QDs to the cysteine residues of tau, as previously reported [30,36,37]. Using this QD-tau, we observed aggregates using an inverted fluorescence microscope and a confocal laser microscope, following previous reports [36,37]. Spatial resolution depends on spatial resolution, and observation is possible at a spatial resolution of approximately 200 nm [38]. As a control, this tau was aggregated at 10 µM in h-tau buffer (50 nM QD-tau, 1 µM heparin as an aggregation promoter, 10 mM DTT as a reducing agent for protein disulfide bonds, under phosphate-buffered saline (PBS) conditions). To mimic phosphorylation conditions, 20 µM tau was incubated at 37 °C for 24 h in 20 mM MOPS (pH 6.8), 1 mM MgCl_2_, and 4 mM ATP. Samples were then aggregated at 10 µM in h-tau buffer: one group retained the phosphorylation buffer, while another group was equilibrated with the phosphorylation buffer using a Zeba™ Spin Desalting Column. Aggregate quantification was performed as previously reported, calculating the amount of tau aggregates as the standard deviation (SD) from the brightness variation per pixel [36].

Observation using an inverted fluorescence microscope (Figure 2A) confirmed that tau aggregates formed within 24 h, with a significant increase in SD value regardless of the presence of the phosphorylation buffer (Figure 2B). Next, the same samples were observed using confocal laser scanning microscopy. ImageJ (version 1.54g, NIH) was used to quantify the density from slice images and the thickness from side-view images of the aggregates. Compared to tau treated with the phosphorylation buffer, tau without the buffer showed a reduced aggregate density and a significant decrease in thickness of approximately 2 μm (Appendix A, Figure 2C,D). This is thought to result from reduced tau concentration due to the 1xPBS used during columnation. This indicates that the desalting process itself may lead to protein loss or structural changes. Measurements showed an approximately 20% reduction in protein concentration. However, since tau prior to aggregation lacks a specific three-dimensional structure, the impact of structural changes caused by using the desalting column is considered minimal. Furthermore, since tau aggregation was confirmed even after column processing [13], this also suggests that the effect of the decrease in concentration is small. Therefore, to equilibrate salts in the buffer and reliably halt phosphorylation, experiments were subsequently conducted using tau without the phosphorylation buffer.

### 2.3. Differences in Aggregate Formation Due to Kinase Variation

Heparin was used in the phosphorylation buffer study to accelerate aggregation time (Figure 2). However, the biological validity of heparin-induced fibrillar structures as a model for tau aggregation induced by a cofactor in vitro has been questioned [14,15]. Therefore, to investigate the formation of physiological filaments, we examined how the morphology and formation process of tau aggregates vary depending on the type of kinase under heparin-free conditions, using tau that had been saturated with phosphorylation by four different kinases. Tau phosphorylated by each kinase was incubated at 37 °C in tau buffer (50 nM QD-tau, 10 mM DTT as a reducing agent for protein disulfide bonds, under PBS conditions) to achieve a concentration of 10 μM. Images were captured every 24 h using an inverted fluorescence microscope.

As a result, only tau phosphorylated by GSK3*β* and MARK4 showed aggregates after 96 h (Figure 3A). Non-p-tau began aggregating around 72 h whereas phosphorylated tau showed a delay of approximately 24 h before aggregation initiation. With heparin present, aggregation reached saturation at around 24 h (Figure 2A,B). In contrast, without heparin, aggregation initiation was delayed by approximately 48 h. Even after 240 h, only non-p-tau, GSK3*β*, and MARK4 aggregates were detected (Figure 3B). Aggregation reached saturation at 192 h, as the SD value remained nearly constant thereafter (Figure 3C). The fact that only GSK3*β* and MARK4 aggregates were detected suggests that these kinases may be involved in the formation of NFTs in AD brains through the phosphorylation of tau.

### 2.4. Differences in Aggregate Shape Due to Kinase Variation

Fluorescence microscopy revealed differences in the formation rate and presence of tau aggregates depending on the type of kinase. To observe the aggregation morphology of phosphorylated tau in greater detail, we examined samples observed by fluorescence microscopy at the 240 h time point using confocal laser scanning microscopy. We quantified the density and thickness of aggregates from the brightness of the obtained slice images using ImageJ (Figure 4A–C, Appendix A).

The results showed no difference in aggregate density between non-p-tau and tau phosphorylated by GSK3*β*, but tau phosphorylated by MARK4 exhibited lower aggregate density (Figure 4B). Furthermore, while MARK4 showed no change in aggregate thickness compared to non-p-tau, GSK3*β* resulted in a significant increase in aggregate thickness. Non-p-tau without heparin exhibited a density approximately seven times higher than that with heparin (Appendix A), while the thickness of the aggregates was confirmed to be approximately one-ninth as thick (Appendix A). This suggests that under conditions closer to physiological conditions, tau aggregates form in a more densely packed manner.

### 2.5. Differences in Tau Filament Structure Due to Kinase Activity

Next, we investigated how the shape of the filaments forming aggregates changed depending on the type of kinase. To observe the phosphorylated tau filaments in detail, we examined samples observed by fluorescence microscopy using transmission electron microscopy (TEM) at the 240 h time point.

The results revealed aggregates composed of linear filaments in non-p-tau. GSK3*β* formed aggregates composed of linear but finely fragmented filaments, suggesting higher rigidity compared to filaments formed by other kinases. MARK4 formed spherical aggregates composed of granular filaments. Furthermore, aggregates composed of wavy filaments were confirmed for p38*α*, for which aggregates were not observable by fluorescence microscopy or confocal laser scanning microscopy. Although aggregates could not be observed for Cdk5/p25, spiral-shaped, undulating fibers were confirmed (Figure 5). These results suggest that tau forming wavy fibers is less likely to form large aggregates observable by fluorescence microscopy. For both GSK3*β* and MARK4, the fibers themselves were finely fragmented. This morphology may promote aggregate formation and deposition within the brain.

### 2.6. Differences in β-Sheet Structures Within Tau Filaments Due to Kinase Variations

Finally, we investigated differences in the proportion of *β*-sheet structures within filaments forming aggregates, depending on the type of kinase. First, ThT was used to confirm the rate of increase and content of *β*-sheet structures within the aggregates. Similarly, we measured phosphorylated tau and non-p-tau, each treated with different kinases, in tau buffer containing 40 μM ThT at 37 °C in 10 min intervals over 240 h.

The results showed a gradual increase in the amount of *β*-sheet structures for MARK4 and p38*α* (Figure 6A). Cdk5/p25 showed a two-step increase at 24 h and 80 h, while GSK3*β* exhibited a sharp increase at around 100 h. Furthermore, the amount of *β*-sheet structures in phosphorylated tau remained at approximately 40% of that in non-p-tau for all kinases (Figure 6A).

Next, circular dichroism (CD) spectroscopy was performed to quantify the structural proportions within each filament. Similarly, phosphorylated tau and non-p-tau, each treated with the respective kinase, were mixed with tau buffer. CD measurements were taken for samples incubated at 37 °C for 0 h and 240 h.

Tau is a protein with a characteristic far-ultraviolet CD spectrum, exhibiting a prominent minimum near 190 nm and showing no significant signal at 220 nm [39]. At 0 h, the spectrum showed the basic tau CD spectrum. At 240 h, the peak shifted to the right, consistent with the presence of a partial *β*-sheet structure, suggesting fibril formation for all kinases (Figure 6B). Furthermore, the CD spectra obtained at 0 h and 240 h were analyzed using BeStSel to calculate the proportion of secondary structures (Figure 6C). For non-p-tau, GSK3*β*, and MARK4, where aggregates were confirmed by fluorescence microscopy, the proportion of β-sheet structures increased significantly. Non-p-tau showed a particularly marked increase in right-twisted *β*-sheet structures, with similar results observed for GSK3*β*. Conversely, Cdk5/p25, MARK4, and p38*α* exhibited a decrease in right-twisted *β*-sheet structures and an increase in left-twisted *β*-sheet structures.

## 3. Discussion

Tau has been extensively studied as a target in AD. In vitro, tau serves as a substrate for numerous kinases, but in vivo, only a few kinases actually phosphorylate tau. Furthermore, few kinases are known to induce NFTs by excessively phosphorylating tau. We have now demonstrated that mouse MBD tau phosphorylated specifically by GSK3*β* and MARK4 forms aggregates. These aggregates exhibit characteristic morphologies, including distinct aggregate and fibril forms, compared to unphosphorylated tau or tau phosphorylated by other kinases (Table 2).

Phos-tag and Western blot analyses confirmed phosphorylation at the primary site, but the possibility of cross-phosphorylation between kinases or non-specific phosphorylation must also be considered. To attribute the observed differences in aggregation specifically to the target kinase, the potential impact of trace impurities from other kinases in the preparation should also be examined. However, given the purity of the kinase used, its influence in this experiment is considered minimal.

Fluorescence microscopy images revealed differences in aggregation rates depending on the kinase involved. Although aggregation initiation was delayed by approximately 24 h compared to unphosphorylated tau, and despite reports that tau phosphorylation inhibits aggregation [4,40], only aggregates of unphosphorylated tau, GSK3*β*, and MARK4 were detectable even after 240 h, indicating aggregation saturation (Figure 4). This suggests that tau phosphorylated by GSK3*β* and MARK4 constitutes a high proportion of the excessively phosphorylated tau forming NFTs. Images captured in 2D using confocal laser microscopy could be reconstructed into 3D, revealing that among the kinases forming aggregates, GSK3*β* formed thicker aggregates compared to MARK4 (Figure 4). It is known that GSK3*β* phosphorylates tau over a broader range compared to MARK4 [41,42]. Differences in three-dimensional structures resulting from varying phosphorylation levels may explain why GSK3*β* formed thicker aggregates. The findings that phosphorylation of GSK3*β* thickens aggregates whereas phosphorylation of MARK4 thins them raise the question of how these different forms correlate with specific pathological roles in AD, such as seeding ability and neurotoxicity. However, cryo-microscopy observations in chronic traumatic encephalopathy and globular Greer tauopathy have reported differences in the fibrillar structure of tau aggregates due to pathology, as well as differences in fibrillar structure within the same disease [29,43,44]. Furthermore, different buffering conditions and the presence of cofactors may result in multiple tau fiber structures originating from the same tau isoform in vitro. Therefore, it is suggested that kinases may also cause differences in pathological effects in AD.

TEM observations were performed to examine the fiber morphology in greater detail, revealing characteristic fiber structures for each kinase. Notably, wavy fibers were observed for Cdk5/p25 and p38*α*, where no aggregates were detected by fluorescence microscopy or confocal laser scanning microscopy. This was supported by quantification of *β*-sheet structure content via the ThT assay, confirming the presence of minute fibers and aggregates with invisible QDs (Figure 6A). Compared to unphosphorylated tau, the initiation of aggregation for GSK3*β* and MARK4 phosphorylated tau was delayed by approximately 24 h. This delay, as indicated by ThT results, showed that while the time to initiate aggregation was delayed for GSK3*β* phosphorylated tau compared to unphosphorylated tau, the overall progression of fiber formation was similar. This suggests that nucleation of phosphorylated tau is more difficult than unphosphorylated tau. In contrast, the fibrillation progression of MARK4 phosphorylated tau increased steadily, suggesting it arises from the formation of a distinct intermediate state requiring a longer accumulation time. Furthermore, non-p-tau showed a particularly marked increase in right-twisted *β*-sheet structures, with GSK3*β* yielding similar results. Conversely, Cdk5/p25, MARK4, and p38*α* showed reduced right-twisted *β*-sheet structures and increased left-twisted *β*-sheet structures. This suggests that increased left-twisted *β*-sheet structures are essential for forming non-linear fibers such as wavy or granular ones (Figure 6C). These findings indicate that phosphorylated tau, particularly those forming wavy fibers, may exhibit reduced aggregate formation or significantly slower aggregate formation rates. In contrast, phosphorylated tau adopting linear or granular fiber morphologies, such as those mediated by GSK3*β* or MARK4, likely form sufficient aggregates and deposit in the brain, even compared to non-p-tau. An increase in right-handed β-sheet structures is thought to enhance stability by forming linear fibers and aggregates. Furthermore, differences in interactions and immunogenicity arise from their three-dimensional structures, necessitating an antibody design that considers the morphology of fibers and aggregates during drug and antibody development.

MARK4 overexpression causes tau to detach from microtubules and leads to NFTs, establishing it as a key kinase in tau phosphorylation. We have now determined that tau phosphorylated by GSK3*β* and MARK4 forms aggregates. While tau in AD brains is phosphorylated at various sites, Cdk5 phosphorylates 9–13 sites. Furthermore, most sites phosphorylated by Cdk5 are also phosphorylated by GSK3*β*. Previously reported phosphorylation sites include Ser202, Thr205, Thr231, Ser235, Ser396, and Ser404 [18]. GSK3*β* is known to phosphorylate most Ser/Thr sites found in tau. Among the sites phosphorylated by Cdk5/p25, GSK3*β* phosphorylates Thr-181, Ser-199, Ser-202, and Thr-205, but not Ser-235 [45,46]. This suggests that, compared to the phosphorylation of tau by Cdk5, phosphorylation of the same sites by GSK3β contributes to NFT formation and aggregation. It has been reported that a mere 20–50% increase in GSK*β* expression in transgenic mouse brains leads to increased phosphorylation at several tau sites [5,47]. This supports its importance as a causative kinase in AD pathogenesis.

Furthermore, MARK4 has been reported to phosphorylate Ser262 and Ser236, which are part of the tau KXGS motif. Phosphorylation at these sites has been shown to significantly reduce tau’s microtubule-binding capacity [5,48,49]. Therefore, MARK4 overexpression is considered a key kinase in tau phosphorylation, as it leads to tau detachment from microtubules and subsequent NFT formation. This suggests that tau detached from microtubules due to excessive MARK4-mediated phosphorylation plays a crucial role in aggregate formation and AD pathogenesis. In this experiment, we examined the effects of phosphorylation by a single kinase. However, p38*α* has been reported to function as a phosphorylation hub, promoting the phosphorylation of GSK3*β*, among other activities [27]. It is necessary to sequentially or simultaneously treat samples with such complex kinases and observe whether their aggregation behavior exhibits synergistic or antagonistic effects. This remains a future challenge.

This study revealed that different types of kinases influence the aggregation morphology and fibrillar structure of tau. However, since NFTs are highly ordered intracellular structures, the observation of different filament forms in vitro raises important questions. How do these kinase-specific filaments assemble into mature NFTs within the cellular environment? Furthermore, whether the efficiency of this process varies depending on the filament requires clarification through future experiments using nerve cells. It is also necessary to analyze the toxicity to cells and determine which kinases phosphorylate and localize the phosphorylated tau deposited in the brain. However, the main challenges in using fluorescent QDs in biological environments are their high toxicity, safety of removal, and biocompatibility issues. Furthermore, the far-ultraviolet region, which is aggressive to biological environments, may induce irreversible photoionization processes. In this experiment, observations using QDs were performed in vitro. When conducting future experiments using cells, employing experimental methods that consider this perspective will likely further clarify the kinases and mechanisms that increase tau toxicity, significantly contributing to the prevention and treatment of AD.

## 4. Materials and Methods

### 4.1. Purification of Mouse MBD Tau

The mouse MBD tau fragment used was prepared according to a previous report [34]. Using the resulting plasmid, bacterial expression and purification of mouse tau MBD were performed according to previous reports [32,33,34]. Briefly, the expression plasmid was transformed into *E. coli* (Rosetta (DE3) pLys, 71403-3CN, Merck, Darmstadt, Germany), and protein expression was induced with 1 mM isopropyl-1-thio-*β*-D-galactopyranoside. The thermostable fraction from each extract was subjected to sequential column chromatography using a UNOsphere™ S (1560113, Bio-Rad Laboratories Inc., Hercules, CA, USA) column and a TOYOPEARL^®^ (Butyl-650, Tosoh Co., Ltd., Tokyo, Japan) column. For UNOsphere™ S and TOYOPEARL^®^ column chromatography, bound proteins were eluted using a gradient of 0–1 M NaCl and 1.2–0 M (NH_4_)_2_SO_4_, followed by dialysis. The purity of the purified proteins was confirmed by SDS-PAGE [50], and their concentration was assessed by the Lowry method [51].

### 4.2. Phosphorylation of Mouse MBD Tau

Mouse MBD tau was mixed with a phosphorylation buffer (20 mM MOPS (pH 6.8), 1 mM MgCl_2_, 4 mM ATP) to achieve a concentration of 20 μM. This mixture was then supplemented with 5 μg/mL Cdk5/p25 (C33-10G, SignalChem Biotech Inc., Richmond, BC, Canada), 20 μg/mL GSK3*β* (G09-10G, SignalChem Biotech Inc.), 20 μg/mL MARK4 (M46-10G, SignalChem Biotech Inc.), and 10 μg/mL p38*α* (M39-10G, SignalChem Biotech Inc.). The purity of each kinase was 80% or higher. A control sample without added kinase was also prepared. Samples were incubated at 37 °C for 24 h to saturate phosphorylation. To equilibrate the phosphate salts in the phosphorylated mouse MBD tau solution with phosphate-buffered saline (PBS), a Zeba™ Spin Desalting Column (89878, Thermo Scientific™, Waltham, MA, USA) was used. First, to wash the desalting column, the column solution was eluted at 4 °C and 1000× *g* for 1 min. This was followed by three sets of the following steps: adding 50 μL of ultrapure water and centrifuging at 4 °C and 1000× *g* for 1 min. Next, 12 μL of phosphorylated tau solution and 3 μL of 1xPBS were added, and desalting was performed at 4 °C and 1000× *g* for 2 min.

### 4.3. Phos-Tag SDS-PAGE Analysis

To confirm the degree of tau phosphorylation, SDS-PAGE was performed on an 8% acrylamide gel to which 5 mM Phos-tag (AAL-107, NARD Institute, Ltd., Hyogo, Japan) was added. The SDS-PAGE gel was quantified using ImageJ software (version 1.54g, National Institutes of Health, Bethesda, MD, USA). Quantification for each individual lane was performed using ImageJ, and the obtained intensities were plotted as bar graphs and line graphs.

### 4.4. Western Blot

Western blotting was performed to confirm the major phosphorylation sites of each kinase. A 20 μM solution of phosphorylated mouse MBD tau was applied to SDS-PAGE. The resulting gel was transferred to a 0.2 μm PVDF membrane (1620714, Bio-Rad Laboratories Inc.) at 53 mA for 1 h. After transfer, the membrane was blocked at room temperature for 30 min in TBST (0.05% Tween 20 (P9416, Sigma-Aldrich Co. LLC, St. Louis, MO, USA), 10% BSA). The membrane was then incubated with the primary antibody, anti-tau rabbit host antibody (T6042, Sigma-Aldrich Co. LLC), rabbit host AT8 (MN1020, Invitrogen, Waltham, MA, USA), Phospho-Tau (Ser262) rabbit host Polyclonal Antibody (44-750G, Invitrogen), and Phospho-Tau (Ser396) Rabbit Polyclonal Antibody (44-752G, Invitrogen) were diluted 1:100, 1:1000, 1:2000, and 1:2000, respectively, and incubated overnight at 4 °C. The membranes were washed three times for 15 min each wash in TBST (0.05% Tween 20) to remove unbound antibodies. The membranes were further treated at room temperature for 1 h with Goat Anti-Rabbit IgG H&L (HRP) (ab205718, Abcam, Cambridge, UK) diluted 1:5000. Unbound antibody was removed by washing three times with TBST for 15 min each wash. The blot was visualized using the EzWestBlue chromogenic reagent (WSE-7140, ATTO Corporation, Tokyo, Japan).

### 4.5. Observation of Mouse MBD Tau Aggregates Using Quantum Dots

The QDot605 (Q21501MP, Invitrogen) used in this study emits fluorescence at 605 nm. In this experiment, QD-tau, in which QD was cross-linked to tau, was used. QD-tau was prepared according to a previous report [16]. Tau solutions were prepared by mixing non p-tau and tau phosphorylated by each kinase with tau buffer (50 nM QD-tau, 10 mM DTT as a reducing agent for protein disulfide bonds, under 1xPBS conditions) to a final concentration of 10 μM. Tau buffer was dispensed into a 1536-well plate (782096, Greiner, Kremsmünster, Austria), centrifuged (5 min, 3700 rpm), and incubated at 37 °C. Subsequently, each well was observed over time using an inverted fluorescence microscope (Nikon C2 Plus, Nikon, Tokyo, Japan) equipped with a CCD camera (DP72, Olympus, Tokyo, Japan) and a confocal laser scanning microscope system (TE2000, Nikon). The density and thickness of aggregates were quantified from the acquired images using ImageJ.

### 4.6. Transmission Electron Microscope Observation

Tau solutions containing non-phosphorylated tau and tau phosphorylated by each kinase were prepared to a concentration of 10 μM. After incubation at 37 °C for 240 h, 10 µL was placed on a colloidal silver-coated membrane (No. U1005, EM Japan K.K., Tokyo, Japan) and incubated at room temperature for 5 min. The grid was washed twice with ultrapure water and treated twice with 0.375% phosphotungstic acid. The grid was thoroughly dried at room temperature for 5 min before observation using a low-acceleration transmission electron microscope (LVEM5, Delong, Montreal, QC, Canada).

### 4.7. Thioflavin T Fluorescence Experiment

Tau solutions containing non-phosphorylated tau and tau phosphorylated by each kinase were prepared to a concentration of 10 μM. ThT was then added to achieve a concentration of 40 μM. Samples were dispensed into a 384-well plate (3540, Corning Inc., Corning, NY, USA), centrifuged (5 min, 3700 rpm), and incubated at 37 °C. Each well was measured at 10 min intervals for 240 h using a microplate reader (SH-9000, Yamato, Tokyo, Japan).

### 4.8. Circular Dichroism Spectroscopy

Changes in the secondary structure of mouse MBD tau during the 24 h aggregation process were monitored using CD spectroscopy. Tau solutions containing non-phosphorylated tau and tau phosphorylated by each kinase were prepared to a concentration of 10 μM. These tau solutions were then incubated at 37 °C for 240 h. Measurements were performed using a spectrophotometer (J-720WI, JASCO Corp., Tokyo, Japan) equipped with a quartz micro-sampling disk cell (MSD-462, JASCO Corp.) and a path length of 0.2 mm. Scanning was performed over the range 185–300 nm with a 1 nm bandwidth, and each sample was measured 10 times. To quantify secondary structure, the BeStSel web server (version 1.3.230210, JASCO Corp.) [36,37,38], which predicts secondary structure content from CD spectra, was used to calculate the proportion of each structure.

## 5. Conclusions

This study demonstrates that phosphorylation mediated by specific kinases determines distinct aggregation pathways for tau protein in AD. We showed that phosphorylation by GSK3*β* and MARK4 promoted the formation of large, dense aggregates, whereas phosphorylation by Cdk5/p25 and p38*α* induced the formation of wavy filaments that suppressed large-scale aggregation. Ser262, a site on Western Blot that significantly affects microtubule binding stability, was phosphorylated only by MARK4. Furthermore, TEM images confirmed the formation of aggregates and fiber morphologies specific to GSK3*β* and MARK4. This suggests that the pathological process could potentially be inhibited not only by monotherapy but also by combination therapy targeting GSK3*β* and MARK4 separately. Furthermore, phosphorylated tau adopting a wavy fibrillar morphology was less prone to aggregate formation, while kinases that increased the amount of right-twisted *β*-sheet structure during fibrillation promoted aggregate formation. By collecting NFTs that accumulate in the brains of AD patients and comparing their aggregate shapes and fiber morphologies, it is conceivable that more accurate subtype classification and personalized treatment based on the phosphorylation profile of tau and the morphology of aggregates within the patient’s brain could be achieved as future therapeutic targets for AD. This direct link between kinase activity and tau aggregate morphology provides a potential mechanistic basis for disease heterogeneity and the establishment of novel therapeutic targets.

## Figures and Tables

**Figure 1 ijms-26-10122-f001:**
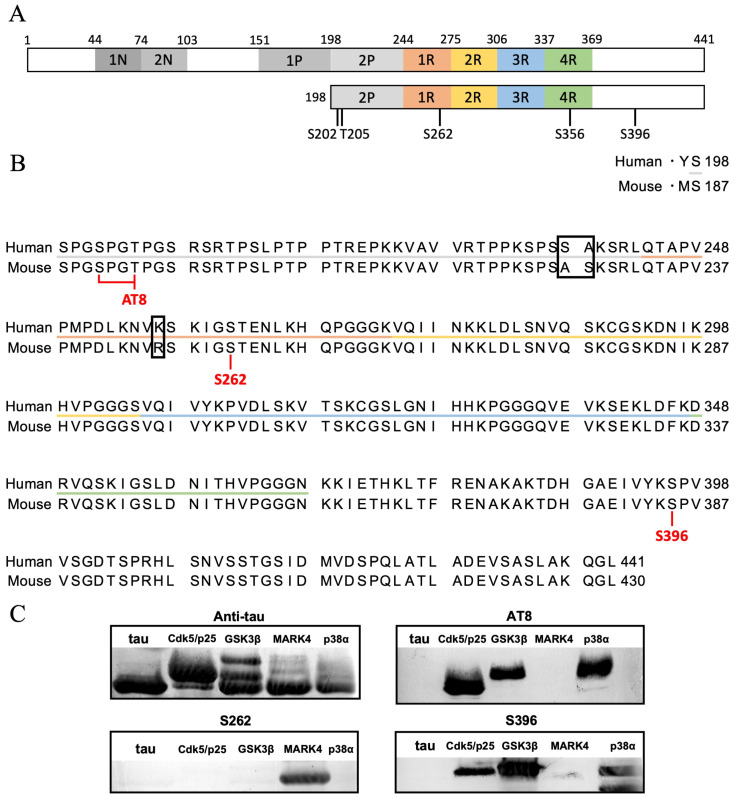
Confirmation of phosphorylation of mouse MBD tau. (**A**) Schematic structure of tau [15]. The structure is classified into the protrusive domain and the MBD. The MBD is further divided into the proline-rich region, repeat region, and tail region. The area used in this experiment is the MBD. The schematic shows the location of the phosphorylation site labeled by a Western blot. (**B**) Comparison of amino acid sequences for mouse MBD tau and human MBD tau used in this study. Orange, yellow, green, and blue indicate each repeat region; gray indicate proline-rich region; black boxes indicate differences in amino acid residues between humans and mice. Red text indicates residues labeled by a Western blot. (**C**) Monochrome images of Western blots labeled with each primary antibody.

**Figure 2 ijms-26-10122-f002:**
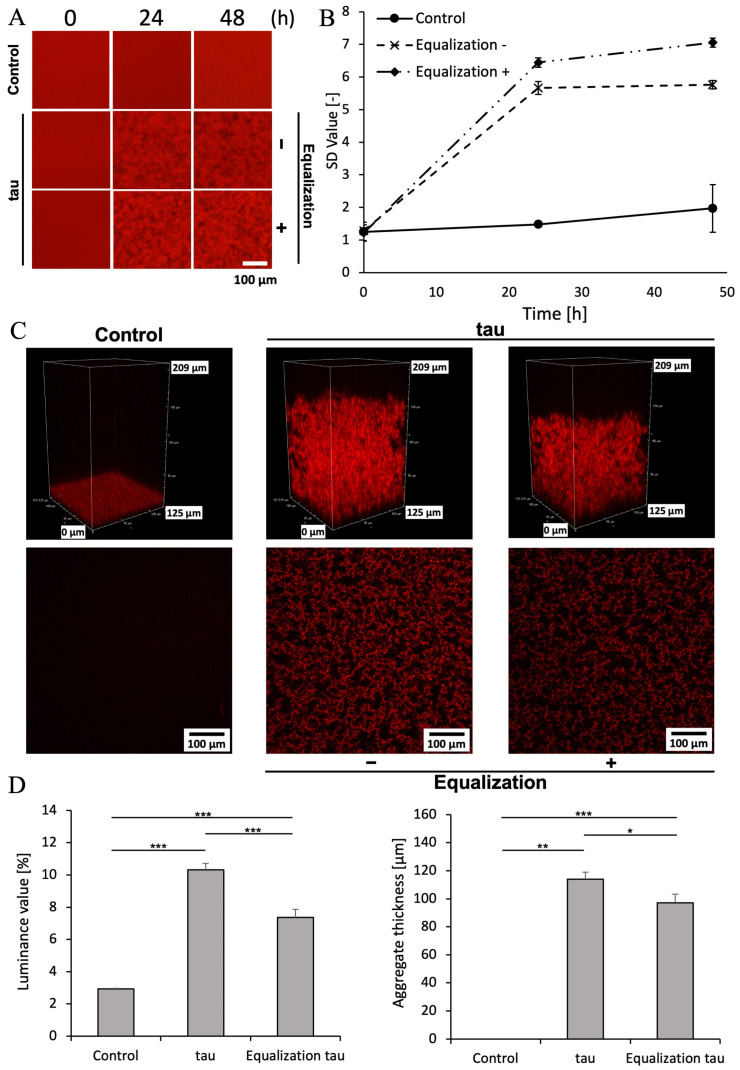
Aggregation of mouse MBD tau under tau aggregation buffer conditions with and without phosphorylation buffer. (**A**) Sequential images of the aggregation process of mouse MBD tau. (**B**) SD values at each observation time point were quantified using ImageJ. Data represent the mean, and error bars indicate SD values calculated from data obtained from three separate experiments. (**C**) Cropped image and slice image of the 3D reconstruction of mouse MBD tau aggregates after incubation for 48 h. (**D**) Aggregate density quantified from slice images and aggregate thickness quantified from side views. Aggregate density was quantified using ImageJ software. Data represent the mean; error bars are derived from images obtained by dividing a single slice image into four sections. Statistical analysis was performed using *t*-tests. *** *p* < 0.001 (*n* = 4). Aggregate thickness was quantified using ImageJ. Data represent the mean; error bars are derived from three points: left, center, and right of the side view. Statistical analysis was performed using a *t*-test. *** *p* < 0.001, ** *p* < 0.01, * *p* < 0.05 (*n* = 3).

**Figure 3 ijms-26-10122-f003:**
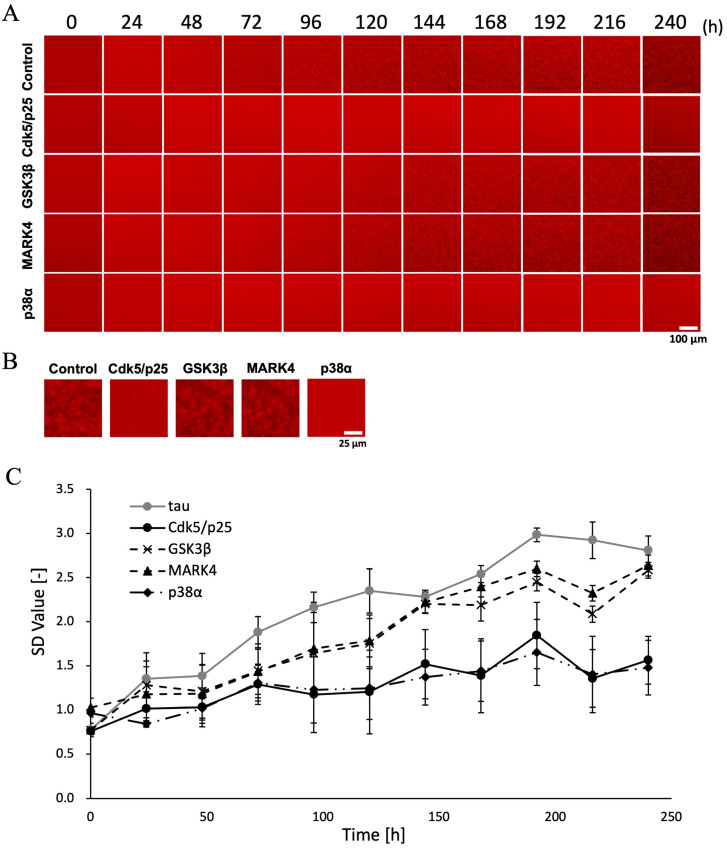
Two-dimensional images of mouse MBD tau aggregation under tau aggregation buffer conditions, showing both unphosphorylated and tau phosphorylated by four different kinases. (**A**) Sequential images of the aggregation process for unphosphorylated and tau phosphorylated by four different kinases. (**B**) Enlarged images at 240 h for each sample in panel (**A**). (**C**) SD values at each observation time point were quantified using ImageJ. Data represent the mean, and error bars indicate SD values calculated from data obtained from three separate experiments.

**Figure 4 ijms-26-10122-f004:**
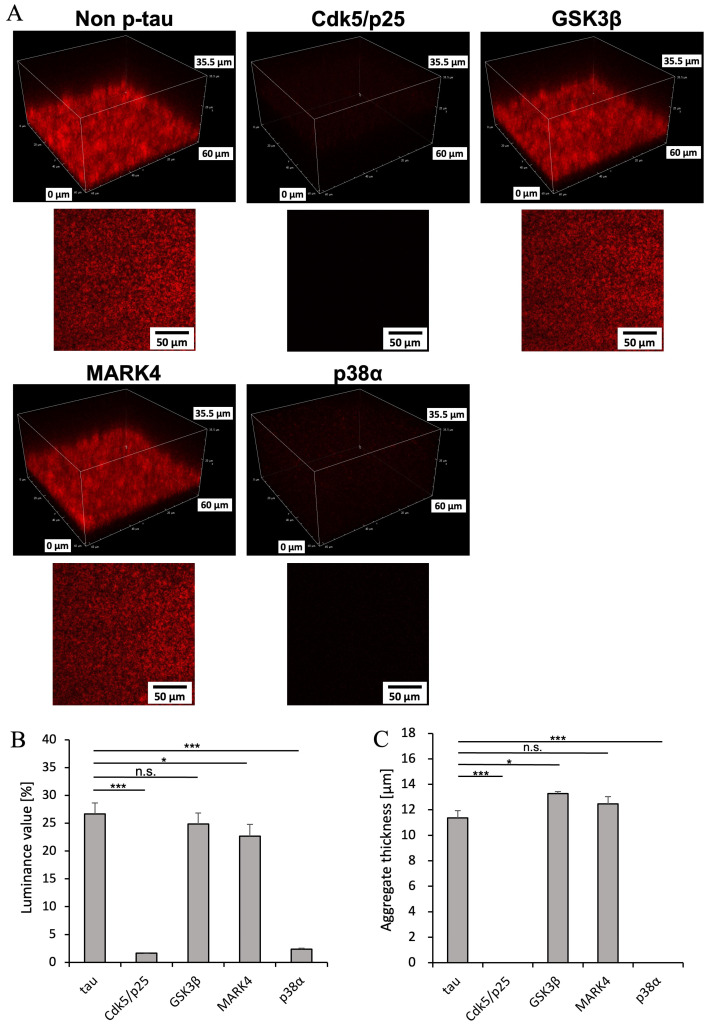
Three-dimensional images of mouse MBD tau aggregates under tau aggregation buffer conditions, showing non-phosphorylation and phosphorylation by four types of kinases. (**A**) Enlarged image and slice image of the 3D reconstructed mouse MBD tau aggregate after 240 h of incubation, as shown in Figure 3. (**B**) Aggregate density quantified from slice images using ImageJ. Data represent the mean; error bars are derived from images obtained by dividing a single slice image into four sections. Statistical analysis was performed using *t*-tests. *** *p* < 0.001, * *p* < 0.05, n.s. indicates non-significant *p*-values (*n* = 4). (**C**) Aggregate thickness quantified from the side view using ImageJ. Data represent the mean; error bars are derived from three points: left, center, and right of the side view. Statistical analysis was performed using *t*-tests. *** *p* < 0.001, * *p* < 0.05, n.s. indicates non-significant *p*-values (*n* = 3).

**Figure 5 ijms-26-10122-f005:**
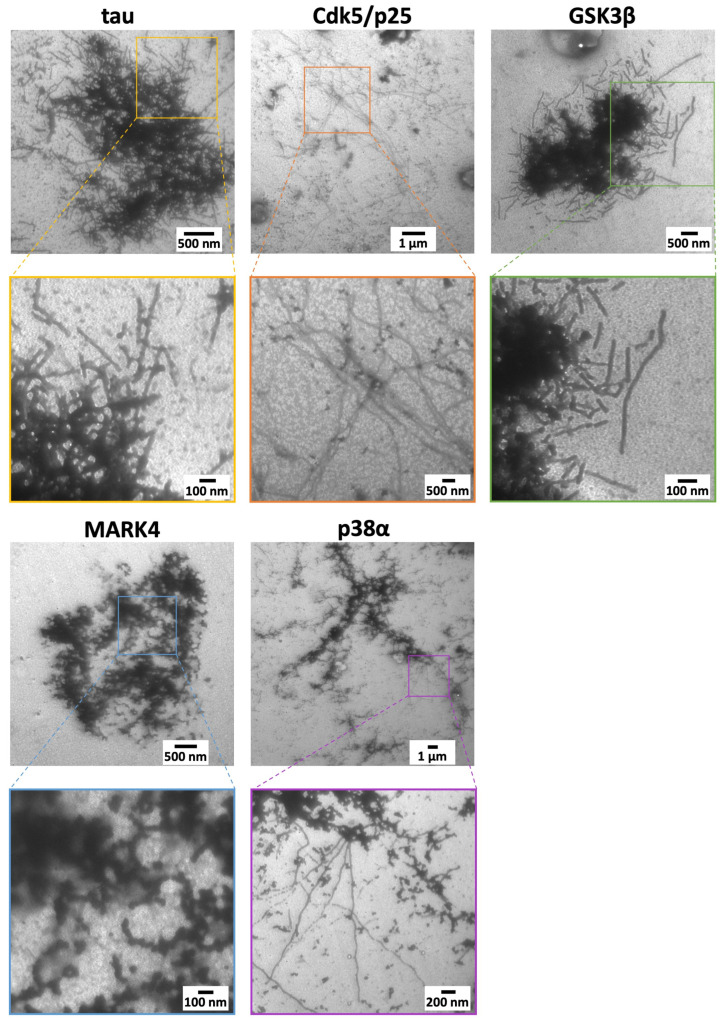
TEM observation of unphosphorylated and four kinase-phosphorylated mouse MBD tau aggregates. Each colored square in the overall image corresponds to a high-resolution TEM image enclosed by a square of the same color.

**Figure 6 ijms-26-10122-f006:**
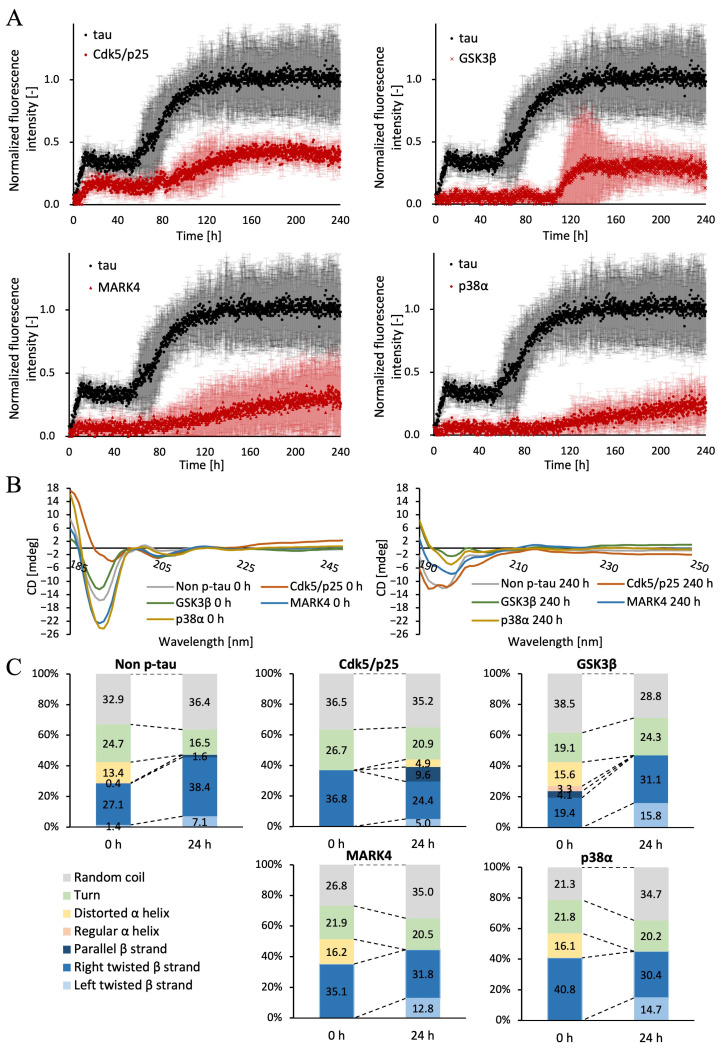
ThT assay and CD spectra of unphosphorylated and four kinase-phosphorylated mouse MBD tau aggregates. (**A**) ThT fluorescence intensity and tau aggregate monitoring of unphosphorylated and four kinase-phosphorylated mouse MBD tau aggregates. Data represent the mean, with error bars indicating SD from three independent experiments. (**B**) CD spectra of each protein before aggregation (**left**) and 240 h after aggregation (**right**). (**C**) Percentage of secondary structural elements calculated using the BeStSel web server.

**Table 1 ijms-26-10122-t001:** Correlation table of primary antibodies reacting with phosphorylated tau by kinase.

Kinase	Anti-Tau	AT8	S262	S396
Non-p-tau	○			
Cdk5/p25	○	○		○
GSK3*β*	○	○		○
MARK4	○		○	
p38*α*	○	○		○

Circles indicate that an antibody reaction was confirmed.

**Table 2 ijms-26-10122-t002:** Characteristics of Aggregates and Fibrous Morphology in Each Observation Results.

Kinase (*n* = 3)	Non-p-Tau	Cdk5/p25	GSK3*β*	MARK4	p38*α*
Luminance value (density) (standard deviation)	11.36 μm (0.57 μm)	0 μm	13.28 μm (0.16 μm)	11.36 μm (0.57 μm)	0 μm
Aggregates thickness (standard deviation)	26.67% (1.96%)	1.63% (0.07%)	24.84% (2.00%)	22.67% (2.14%)	2.35% (0.17%)
Shape of aggregates	Mesh-like	Only fibers confirmed	A collection of fragments	Thick aggregates	Radial
Shape of fibers	Straight	Wave	Rigid torsional straight	Granular	Twisted wave
Increased fibers	2-step increase	2-step increase	Sharp rise in the middle	Gradual increase	Gradual increase
Increased *β*-sheet structure	Right-twisted *β*-strand	Left-twisted *β*-strand	Right-twisted *β*-strand	Right-twisted *β*-strand	Left-twisted *β*-strand

## Data Availability

The original contributions presented in the study are included in the article, further inquiries can be directed to the corresponding author.

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
