# Peer review of "Aggregation Characteristics of Tau Phosphorylated by Various Kinases as Observed by Quantum Dot Fluorescence Imaging"

_ijms, 2025, doi:10.3390/ijms262010122_

Round 1
Reviewer 1 Report
Comments and Suggestions for Authors
I reviewed the manuscript entitled Aggregation characteristics of tau phosphorylated by various kinases as observed by Quantum Dot Fluorescence Imaging.
I agree to accept this manuscript after major revision.
1) amyloid-β (Aβ), p38α, Greek letters need to be italicized, please check and modify the entire text.
2) Tau, a microtubule-associated protein, belongs to the MAPs family, What is the full name of MAPs? When it first appears, the full name should appear and the abbreviation should be in parentheses. Abbreviations are only necessary if they appear three or more times. Because too many abbreviations can confuse readers. But I checked and found that the full text of MAPs only appears once. Therefore, simply use the full name.
3) The following four kinases are commonly cited as phosphorylating tau. Phosphorylation of tau by Cdk5/p35 is considered a physiological process, while phosphorylation by Cdk5/p25 is regarded as a pathological process. Cdk/p25 accumulates in AD brains and is known to possess higher tau phosphorylation capacity compared to Cdk5 activated by p35. Cdk/p25 accumulates should change to Cdk5/p25 accumulates.
4) Glycogen synthase kinase β (GSK3β) binds to microtubules, Glycogen synthase kinase β should change to Glycogen synthase kinase 3β.
5) Furthermore, numerous reports have documented that tau is phosphorylated by various kinases [7]. This statement requires at least two references to be cited.
6) 20 μg/ml GSK3β, 20 μg/ml MARK4, 10 μg/ml p38α, μg/ml should change to μg/mL. 12 μl should change to 12 μL. Check and modify similar issues throughout the entire text.
7) Figure 2. indicates non-significant p-values (n = 3). When it comes to statistics, p should be italicized; n should also be italicized. Check and modify similar issues throughout the entire text. Significant markers with p<0.1 appear multiple times in the text. In the field of life sciences, the significance threshold is usually set at p<0.05. Do author need clearer explanations or more cautious conclusion statements when using the standard of p<0.1?
8) 2.5. Differences in tau filament structure due to kinase activity; the first letter of each actual word in the secondary title needs to be capitalized. Please check and modify similar issues throughout the entire text.
9) Table 2. The number of decimal places should be consistent. For example, Aggregates thickness, 26.7%, 1.63%, 24.8%, 22.7%, 2.35%, please check and modify similar issues.
10) While Phos-tag and Western blot analyses confirmed phosphorylation at key sites, the possibility of cross-phosphorylation among kinases or non-specific phosphorylation must be considered. To firmly attribute the observed differences in aggregation specifically to the target kinases, the study should address the potential influence of trace kinase impurities in the preparations.
11) In Figure 2, a desalination column was used to eliminate the influence of phosphorylation buffer. But the desalination process itself may lead to protein loss or conformational changes. Is there any data to prove that the natural conformation and aggregation rate of tau protein treated with desalination columns are not significantly different from those of the untreated group?
12) The finding that GSK3β phosphorylation produces thicker aggregates, whereas MARK4 phosphorylation yields less dense ones, raises the question of how these distinct morphologies are correlated with their specific pathological roles in AD, such as seeding potency and neuronal toxicity. Furthermore, is this morphological dichotomy supported by existing literature suggesting that different aggregate structures possess varying pathological effects?
13) CD spectra show that different kinases lead to an increase in left-handed or right-handed β - folding structures. What potential impact does the chiral difference of this secondary structure have on the stability of fibers, their interactions with other proteins, and immunogenicity? This is a very novel discovery that deserves further exploration.
14) Compared with non phosphorylated tau, the aggregation initiation of GSK3β and MARK4 phosphorylated tau was delayed by about 24 h. Is this delay due to the difficulty of phosphorylated tau nucleation or the formation of different intermediate states that require longer accumulation time?
15) In the AD brain, multiple kinases work together. Has this study considered sequentially or jointly treating tau with two kinases (such as GSK3β and MARK4) to observe whether their aggregation behavior exhibits synergistic or antagonistic effects?
16) However, as neurofibrillary tangles (NFTs) are highly ordered intracellular structures, the observations of distinct filament morphologies in vitro raise critical questions. How do these kinase-specific filaments subsequently assemble into mature NFTs within a cellular environment, and does the efficiency of this process differ among them?
17) The conclusion suggests that GSK3 β and MARK4 are potential therapeutic targets. Does this mean that combination therapy for AD (simultaneously inhibiting GSK3 β and MARK4) will be more effective than monotherapy? Or is inhibiting one of them enough to interfere with the pathological process?
18) The finding that different kinases lead to distinct morphological forms of tau aggregates raises a question: could this correspond to different subtypes existing among Alzheimer's disease patients? Is it possible to achieve more precise subtyping and personalized treatment based on the tau phosphorylation profile or aggregate morphology in a patient's brain?
19) References, all references are missing DOI numbers, please add them.
20) This study reveals that specific kinase-mediated phosphorylation dictates distinct aggregation pathways of tau protein in Alzheimer's disease. We demonstrate that phosphorylation by GSK3β and MARK4 promotes large, dense aggregates, whereas Cdk5/p25 and p38α induce wavy filaments that resist large-scale aggregation. This direct link between kinase activity and tau aggregate morphology provides a potential mechanistic basis for disease heterogeneity and novel therapeutic targeting.
21) The biggest problem with this study is many details need to be modified and improved. Too many issues can make people feel that the author's attitude is not rigorous.The author must take them seriously and make necessary revisions.
Reviewer 2 Report
Comments and Suggestions for Authors
This is an interesting study in which authors examined how different kinases phosphorylate tau protein and influence its aggregation into distinct fibrillar morphologies, which indicated that GSK3β and MARK4 phosphorylation promote robust aggregate formation. Results of this study may contribute in explanation of kinase-specific roles in tau pathology and their potential effect on NFT formation in Alzheimer’s disease. I recommend publishing this article after some corrections.
- It would be beneficial to improve the form of the Abstract since it is too extensive including more than enough methodological and mechanistic details that belong in the section „Introduction“ rather than the Abstract. While these points are valuable, they overwhelm the reader instead of providing a quick overview.
- Three letter abbreviations for amino acids considered in line 51 should be explained as they are the first time mentioned in the text. In addition, some of the abbreviations are not explained anywhere in the text, such as Cyclin-dependent kinase 5 or a specific sequence motif KXGS found within the tau protein.
- On the basis of the statements from lines 107-111 it is not clear if the authors investigated differences in aggregate shape and formation process of four (Cdk5/p25, GSK3β, MARK4, p38α) or two (GSK3β and MARK4) kinases.
- Why is the y-axis shorter than the label indicating the p-values in Figure 2D?
- It would be very useful if some states are supported by corresponding references such as „In vitro, tau serves as a substrate for numerous kinases, but in vivo, only a few kinases actually phosphorylate tau. Furthermore, few kinases are known to induce NFTs by excessively phosphorylating tau.“
- It would be interesting to add some comment on the possibilities of bridging these in vitro aggregation patterns with functional outcomes in neuronal systems and, regarding future perspectives, suggest which experimental approaches, would be the best choice to determine whether the observed kinase-specific tau aggregates are indeed drivers of Alzheimer's disease pathology?

Reviewer 3 Report
Comments and Suggestions for Authors
Dear Authors,
thank you for the provided scientific work.
In this study, the Authors demonstrated abnormal tau phosphorylation and the direct process of its aggregation, characteristic of Alzheimer's disease. Subsequently, a comparative analysis of the morphology and formation of phosphorylated tau aggregates produced by four kinases: Cdk5/p25, GSK3β, MARK4, and p38α was conducted. The Authors report that, using quantum dots for two-dimensional and three-dimensional structural analysis, the presence of tau aggregates was confirmed in unphosphorylated tau, as well as in tau phosphorylated by GSK3β and MARK4. It is expected that the information obtained will contribute to elucidating the pathogenesis of Alzheimer's disease and the development of new therapeutic strategies.
Comments:
- The title of the article focuses on an innovative approach to using quantum dots to study structural features during various formation processes. In general, quantum dots are color centers that exhibit bright luminescence when excited at a certain wavelength. Lines 286-287 contain information about the far-ultraviolet emission band of the tau protein. However, there is no information about the quantum dots themselves, limited to lines 155-156. I understand that the article's space may be limited, and the Authors cited three previous references. However, I believe that if the Authors include this method in the title, then comprehensive information about the methodology (quantum dot type, optical properties, etc.) should be provided in the current paper.
- What is the spatial resolution achieved for quantum dots?
- The main challenges of using fluorescent quantum dots in biological environments are their high toxicity, safe removal, and the question of biocompatibility. Furthermore, the Authors mention the far ultraviolet region, which is an aggressive range for biological environments, leading to irreversible photoionization processes. Commentary is required.
- The figures in the article have poor resolution.
Reviewer 4 Report
Comments and Suggestions for Authors
NICE AND INTRESTING WORK
ADD MORE AND MORE DETAILED CONCLUSIONS-MENTION THE INNOVATIONS OF THIS STUDY-THE CONCLUSIONS HAVE TO BE ADEQUATE TO THE LENGTH OF THE TEXT
RE CHECK THE ENGLISH LANGUAGE USED
REFERENCES ARE OK BUT YOY HAVE TO ADD MORE AND MORE RECENT REFRENCES-THE REFERENCES HAVE TO BE BE ADEQUATE TO THE LENGTH OF THE TEXT
IN MATERIAL AND METHODS SECTIONS MINIMIZE THE TECHNICAL CHARATERISTICS OF THE METHODS USED
ENLARGE AND PUT COLOUR IN THE TABLE 2 IN ORDER TO AMELIORATE THE TEXT
ENLARGE FIGURE 6 IN ORDER TO AMELIORATE THE TEXT
FIGURE 5 PUT COLOURS IF POSSIBLE
FIGUR 4 ENALARGE B AND C SECTION AN CLARIFY MORE THE STATISTICS USED
ENLARGE FICURE 3 C SECTION IN ORDER TO AMELIORATE THE TEXT
FIGURE 2 ENLARGE B AND D SECTION
AND VCLARIFY BETTER THE STATISTICS USED
ENLARGE TABLE 1
Comments on the Quality of English LanguageThe English could be improved to more clearly express the research.
Round 2
Reviewer 1 Report
Comments and Suggestions for Authors
The author has made the necessary modifications and explanations as per my request, therefore I agree to accept it in its current form.
Reviewer 3 Report
Comments and Suggestions for Authors
The article has been substantially revised. I am satisfied with the reasoned responses to my comments. The manuscript can be accepted for further processing.
Reviewer 4 Report
Comments and Suggestions for Authors
nice work
improvment in all the sections
compliments
Comments on the Quality of English LanguageThe English are improved-compliments